# Religious Experiences in the Context of Bipolar Disorder: Serious Pathology and/or Genuine Spirituality? A Narrative Review against the Background of the Literature about Bipolar Disorder and Religion

Eva Ouwehand 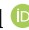

Altrecht GGZ, 3512 PG Utrecht, The Netherlands; e.ouwehand@altrecht.nl

**Abstract:** Literature about bipolar disorder and religion is scarce and primarily encompasses studies with a quantitative design. Results of such studies do not lead to unambiguous conclusions about the relation between bipolar disorder and religion that could be applied in clinical practice. The main focus of this article will be on the domain of religious experiences/religious delusions and hallucinations as explored in two recent PhD studies regarding mixed methods and qualitative research, conducted in the Netherlands and in Canada. In the narrative review of the two studies, the occurrence of different types of religious experiences and various explanatory models of patients to interpret them are presented. The interpretation of religious experiences, often related to mania, proves to be an intense quest, and often a struggle for many patients, whereby fluctuations in mood, course of the illness, religious or philosophical background, and the reactions of relatives and mental health professionals all play a role. Patients combine various explanatory models, both medical and religious/cultural, to interpret their experiences and these may fluctuate over the years. The two studies are placed in the context of literature about bipolar disorder and various aspects of religion to date. Finally, the challenges for future research and the implications for clinical practice will be outlined.

**Keywords:** bipolar disorder; psychopathology; religious experiences; explanatory models; religion; spirituality

## 1. Introduction

Bipolar disorder in relation to religion or spirituality is an understudied subject. In this article, the focus will be on the relation between bipolar disorder and experiences perceived as religious or spiritual by persons with this diagnosis, reviewing two recent dissertations on this topic (Ouwehand 2020, The Netherlands; Van der Tempel 2022, Canada). Both studies contain qualitative research from a patient perspective and provide insight into the personal interpretation processes of experiences that, from a psychiatric perspective, are often viewed as pathological.

These studies differ from the majority of studies in the field, which, for the most part, have a quantitative design and explore the relationship between various religious variables and health outcomes or severity of the illness. Because the literature is sparse, in this introduction we will first outline the general scientific state of affairs regarding bipolar disorder and religion or spirituality. Four review studies are available, all from the past 12 years, altogether containing 19 studies (Pesut et al. 2011; De Fazio et al. 2015; Koenig 2018; Jackson et al. 2022). A variety of religious variables are used in predominantly quantitative research designs, pointing to different conclusions.

After this broad summary of research to date, in the section we will elaborate more on the two dissertations regarding religious experiences and bipolar disorder in Section 2, to be followed by some conclusions for future direction, both in research and in clinical

practice, in Section 3. An important assumption in this article is that religious phenomena cannot be analysed independently from the cultural context in which they occur. The geographical situation of the studies is important for the interpretation of the results, as the role religion plays in societies and individual lives varies greatly. The studies about religious experiences and bipolar disorder, discussed in Section 2, are clear examples of the importance of considering the—in this case—secularized context, wherein a religious phenomenon is studied. Religion, spirituality, and secularism are part of the broader culture in which they flourish and, especially in quantitative studies, this context often does not receive enough attention. The country where studies are conducted is therefore mentioned in all cited research, when relevant.

### 1.1. Religious Coping and Intrinsic Religiosity

Research of the relationship between religiosity and bipolar disorder does not unambiguously point in one direction. In some studies, religious variables such as church attendance or religious affiliation seem to be a protective factor against the development of mania (Baetz et al. 2006, Canada), or point to a later onset of illness and hospital admission (Dervic et al. 2011, USA). However, Mizuno and colleagues (Mizuno et al. 2018, Austria, Japan) found a moderate association between higher levels of religiosity (church attendance, religious activities) and residual symptoms of mania in bipolar l patients. Baetz and colleagues (Baetz et al. 2006), in the above-mentioned population-based study on community health in Canada, reported that higher importance attached to spiritual values of patients (e.g., meaning of life, coping with daily difficulties, insight in life problems) was associated with higher frequencies of depression and mania.

Religious coping can be an important way of dealing with the negative effects of the illness, even when they are severe. A religious frame of reference may give a name to and understanding of human suffering and can provide access to religious or spiritual resources. It was an important way to manage the illness, both for patients and their relatives, in a qualitative Israeli study of coping strategies of patients and partners (Granek et al. 2018). According to a Turkish study (Çuhadar et al. 2015), religious coping most frequently occurred among outpatients with an adaptive coping attitude focused on healthy behavior, irrespective of psychotic or manic symptoms. Intrinsic religiosity, which means that faith or spirituality have an intrinsic value for a person and are not only means to an end (for example being part of a community), is used as a religious variable in studies as well. As some studies among bipolar patients suggest (mainly in non-western, less secularized countries), positive religious coping and intrinsic religiosity are positively associated with fewer depressive symptoms and a higher quality of life (Stroppa et al. 2018, Brazil; Stroppa and Moreira-Almeida 2013, Brazil), with less suicidality (Caribé et al. 2015, Brazil), or with a better recovery (Grover et al. 2016b, India). Other studies, on the other hand, do not find any relationship between intrinsic religiosity and symptoms of the illness (AbdelGawad et al. 2017, USA), or, on the contrary, point to a relation between religious variables and an increase or severity of illness symptoms, for example in episodes with mixed symptoms, which are usually more serious in nature than single manic or depressive episodes (Cruz et al. 2010, USA).

No clear picture can be drawn of the importance that people with bipolar disorder in secularized countries attach to religiosity or spirituality. In a Swiss study (Huguelet et al. 2016), intrinsic religiosity appeared to play a more prominent role in the lives of patients with schizophrenia (41%) than in the lives of bipolar patients (6%). In contrast to these results, other studies report the importance of faith or spirituality for bipolar patients (Mitchell and Romans 2003, New Zealand; Ouwehand 2020, The Netherlands). In the former study, patients saw a direct link between their religiosity and illness management, especially among groups that had been searching for spiritual healing (evangelical Christians and Maori). However, this view did not lead to better mental health in the preceding five years. This finding indicates that the level of religious coping can be related to someone's specific cultural background. That was evident, for instance, in the study of Pollack

and colleagues (Pollack et al. 2000, USA), in which Afro-Americans appeared to have more religious resources at their disposal to cope with their illness than white Americans.

Negative religious coping, which refers to people who seriously doubt their faith, feel abandoned or judged by God, or are angry with God, is another understudied aspect in relation to bipolar disorder. Stroppa et al. (2018, Brazil) found that negative religious coping was related to a lower quality of life and more manic symptoms. In the study by Mitchell and Romans (2003, New Zealand), 40% of the participants were disappointed in their faith due to their illness. This could indicate negative religious coping as well.

*1.2. Fasting*

A specific topic, namely the influence of fasting on symptoms or development of bipolar disorder, is addressed in studies, conducted in countries with Islamic majorities. A review study (Eddahby et al. 2014, Morocco) of French and English research from 1970 to 2011 concluded that advising patients with bipolar disorder in regard to Ramadan was difficult, because findings related to the effect of fasting on the illness are contradictory. Not only changed eating patterns but also variation in social rhythm may influence the course of the illness. According to Farooq and colleagues (Farooq et al. 2010, Pakistan), fasting did not have any effect on pharmacological treatment (lithium levels, negative side effects, toxicity), while depressive and (hypo)manic symptoms decreased during Ramadan. Other studies report symptom relapse, despite unchanging lithium levels (Kadri et al. 2000, Morocco). A more recent explorative study in Tunisia found that two thirds of patients fasted regularly before the onset of bipolar disorder, but more than half of them stopped fasting after their diagnosis, mainly to ensure therapeutic compliance (Mejri et al. 2023, Tunesia). Because fasting is a practice in many religious traditions, the topic deserves attention in clinical practice and in future research.

*1.3. Religious Experiences, Religious Delusions and Hallucinations*

The area between religious or spiritual experiences and delusions and hallucinations in relation to bipolar disorder is usually studied either from the perspective of religious experiences and interpretations thereof, or from the bio-medical perspective, which often entails the view that such experiences are psychopathological symptoms of bipolar disorder. Scientific literature on the topic is scarce. Two older American studies, conducted in areas with a predominantly Christian population, compared the prevalence of religious experiences of bipolar patients with other groups. Gallemore and colleagues (Gallemore et al. 1969) studied conversion and salvation experiences of bipolar patients (n = 62) and found a prevalence of 52% in this group, against 20% in a healthy control group. However, only in a few cases were the experiences related to a manic episode. Kroll and Sheehan (1989) reported a prevalence of 55% of 'personal religious experiences' in an inpatient group of bipolar patients (n = 11) against 35% in the general population.

Studies that start from a medical viewpoint estimate the prevalence of religious delusions in mania to be 15–33% in the United States (Appelbaum et al. 1999; Koenig 2009). In India, Grover and colleagues (Grover et al. 2016a) estimated religious psychopathology to be apparent in 38% of their sample of patients with bipolar disorder, whereof two thirds had a Hindu affiliation and one third Sikh. Critique of an either religious or pathological approach is articulated by Cook (2015), who, in a review study into religious psychopathology, highlights the lack of an agreed scientific definition of what healthy or pathological religiosity exactly encompasses. Another point of criticism is the interconnectedness of both religious experiences and religious psychopathology with the wider cultural context in which they occur (Luhrmann 2011). This makes comparison of figures of a concept such as 'religious delusion' between countries a sensitive endeavor.

Apart from the problem of scientific clarity in research, patients themselves struggle with the question of the authenticity or the pathology of their experiences (Michalak et al. 2006, Canada) and with the tension between a religious and a medical explanatory model for such experiences (Mitchell and Romans 2003, New Zealand; Stroppa and Moreira-

Almeida 2013, Brazil). This tension may have consequences for the treatment relationship, and will probably be expressed differently in different cultural contexts. A multidisciplinary approach, both in clinical practice as in research on the complex relationships between religion and mental health, could be fruitful.

## 2. Interpretation of Religious Experiences Related to Bipolar Disorder: Two Studies

Recently, the topic of occurrence and interpretation of religious experiences in patients with bipolar disorder has been explored in greater depth in two dissertations (Ouwehand 2020, The Netherlands; Van der Tempel 2022, Canada); these will be presented in this section and complement each other. Both studies contain qualitative research and give insight into the patient perspective on both descriptions of religious experiences and the interpretation process thereof in retrospect. The former study was of mixed methods design. In the qualitative part of the study, persons (n = 34) with a more than average interest in faith and spirituality participated. The quantitative part of the study, conducted in a specialized outpatient center for bipolar disorder, moderated the results of the qualitative part of the study. The latter study, with a more modest qualitative design (n = 11), included, interestingly, only atheists, of whom nine adopted an agnostic or spiritual/religious worldview after their experiences during (hypo)mania (Van der Tempel 2022). The study used the variation in religious experiences found in the former study (Ouwehand et al. 2019a) as a template for analysis of participants' narratives regarding their religious experiences.

### 2.1. Theoretical Background

The first study (Ouwehand 2020), conducted in The Netherlands, took a religious-studies perspective as a theoretical basis. It combined qualitative interviews about religious experiences and interpretations thereof with quantitative research (n = 196) in a group of out-patients of a specialized mental health care facility for bipolar disorder in The Netherlands. The description and interpretation of religious experiences was first studied phenomenologically from a patient perspective in qualitative interviews. Such experiences are embedded in the secularized and pluralistic contemporary Dutch society, in which concepts such as religion and spirituality are highly subject to change and will therefore evoke a variety of associations in participants in a study (the emic perspective). A decrease of institutional religion and religious traditions and an increase in interest in personal, privatized, noninstitutional, and experiential aspects of religion is one of the trends in secularizing western societies (Possamai 2005; Streib and Hood 2016). Following Streib and Hood, Ouwehand takes the concept of 'spirituality' as one of the various forms that religion can take (the etic perspective), irrespective of emic associations with the terms spirituality and religion. In the study of Van der Tempel (2022), only persons who identified as atheist at the time of their first religious experiences were included, implying, in this study, that participants did not believe in the existence of god(s) or supernatural/transcendent beings or realities.

The variety of meanings people attribute to the concepts of religion and spirituality has prompted the use of the self-definition of persons as religious and/or spiritual, or neither religious nor spiritual, in contemporary research in the sociology of religion. Self-definition as a variable is controversial, because it is not clear whether the non-religious use the term 'neither religious nor spiritual' to describe themselves. Are they agnostic, atheistic, or perhaps New Age believers? Van der Tempel (2022) asks. Therefore, a self-definition approach needs to be coupled with other variables such as religious belief, behavior, or religious experiences (Berghuijs et al. 2013, The Netherlands) or combined with qualitative research. A combination of methods and instruments may, in its results, reflect the contemporary religious and spiritual landscape in western, secularizing populations more adequately than using religious variables such as church attendance or prayer only in relation to mental health. In the quantitative part of the study, self-definition was used as a religious variable next to regularly used research instruments in the field of religion and

health for quantitative studies, such as the religious coping scale (RCOPE, Pargament 1999) or the Duke University Religions Index (DUREL, Koenig and Büssing 2010).

Both studies (Ouwehand 2020; Van der Tempel 2022) focused on post-hoc meaning-making processes of religious experiences of individuals with bipolar disorder and employed a constructivist approach. Ouwehand applies the term 'explanatory models' for illness experiences, coined by psychiatrist and cultural anthropologist Arthur Kleinman (1988, 1991), to religious experiences related to bipolar disorder. It refers first to how patients and their relatives explain and cope with disease symptoms. Second the term refers to the biomedical model, which transforms a person to a 'patient' with a diagnosis. It encompasses the theoretical framework behind diagnoses, including Nosology, epidemiology, and evidence-based treatment interventions, as well. Third, the term 'explanatory model' refers to broader cultural notions of illness and health, often apparent in popular, non-scientific theories. All three levels of interpretation are viewed as social constructions by Kleinman, and these interfere with one another, according to this author. Personal explanatory models often have a religious or spiritual content and are influenced by religious views on illness and health in society. Kleinman states that the treatment relationship benefits from addressing the different explanatory models during treatment because they may reinforce one another, but can be contradictory as well. He has mainly researched non-western cultures, where medical and scientific knowledge is much less diffused throughout society.

In a Western context, with high-level medical facilities and integration of scientific knowledge within the educational system, as in The Netherlands, cultural notions about illness and health may refer to both popular scientific medical knowledge and to various alternative or complementary therapies. The latter often contain religious or spiritual components that explain illness and health in a broader spiritual framework and help people to make sense of what is happening to them when they experience psychological stress or disruption (Hoffer 2012). 'Healing' and 'spiritual growth' are important themes in new forms of spirituality or New Age religion (Hanegraaff 1996; Heelas et al. 2005).

*2.2. Summarised Findings of the Two Qualitative Studies*

2.2.1. Religious Experiences

In 34 semi-structured interviews with recovered Dutch bipolar patients, religious and spiritual experiences, interpretations thereof, and their cultural influence on the lives of the participants were explored (Ouwehand et al. 2018, 2019b). A variation of experiences, perceived as religious or spiritual, was found, mostly positive and often related to mania (Ouwehand et al. 2018). The experiences were of both horizontal and vertical transcendence (Goodenough 2001; Streib and Hood 2013). An example of the former are experiences comparable with the peak experiences of Maslow (1959), like 'an intense experience of happiness, love, peace, beauty or freedom', or 'an experience of meaningful synchronicity', when people perceive what happens not as incidents but as events occurring with a purpose. An example of the latter is the experience of the presence of God or of a supernatural reality, "an intense closeness of God, as if He is sitting in the room" (Woman, Protestant, Ouwehand 2020, p. 58). Other experiences were mystical experiences of unity (both horizontal and vertical), for example: "The feeling that I am one with everything, you feel a kind of oneness between people and basically it is in everything you see—that there is no separation anymore" (Man, New Spirituality, Ouwehand 2020, p. 48), of vocation or mission, paranormal experiences, such as contact with deceased persons or out-of-the-body experiences, apparitions and voices, symbolic images or visions and perceiving oneself as an important religious person.

Van der Tempel (2022) found some similar categories in his sample of atheist patients: 'divine identity' (being (like) god) and/or 'having a divine mission'; 'divine intervention or synchronicity'; 'connection and transcendence', similar to experiences of oneness or mystical experiences in the study of Ouwehand and colleagues; and 'revelation and clarity', comprising insights in the nature of reality with full clarity. An example of an

experience of 'connection and transcendence', comparable to mystical experiences, was the following description:

> "I just felt like I was connected to everything in the universe. And it was mostly when I was outside in nature. It was every single thing, like the leaves and the clouds and whatever. I felt like I could see connections between everything. And I was a part of it too. It was just this beautiful experience of being part of a world that's so alive and vibrant and just beautiful, right?". (Woman, Van der Tempel 2022, pp. 85/86)

Negative religious experiences were far less mentioned, but they were present as well in both studies. In the study of Ouwehand and colleagues (Ouwehand et al. 2018), they occurred in relation to both mania and depression. An example of a negative experience was reported by a Roman Catholic man: "It's as if I don't exist, as though I'm just a shell and in another dimension. That's a frightening feeling" (Ouwehand 2020, p. 58). During depression, faith or spirituality was absent in more than half of the interviews, or the divine was perceived as being absent. Themes such as guilt and punishment, religious doubt, or the presence of evil were reported by a third of the 34 interviewees in the Dutch study, as well as the theme of trust and confidence despite despair, for example "I thought: what if I sinned against the Holy Spirit? It was as if I was at the edge of Purgatory" (Woman, New Spirituality, Ouwehand 2020, p. 58). In the study of Van der Tempel (2022) some participants reported stress and doubt about what was happening to them during the religious experiences they had. Another finding in his study was that the occurrence of religious experiences was often preceded by various mental health difficulties, such as long-standing depression, stressful life transitions, grief, and internal conflict, according to participants.

### 2.2.2. The 'both/and' Explanatory Model and Lasting Influence of Religious Experiences

By far the largest group in the Dutch qualitative study attributed a religious or spiritual meaning to their experiences and considered them as having a positive influence on their lives (Ouwehand et al. 2019b). They described this influence in terms of a process of personal transformation, insight into oneself, into the purpose of life, into the nature of reality, or into the nature of good and evil. At the same time, they were aware of the disruptive aspects of their experiences. They mentioned, for example, being overly preoccupied with them, to the extent of neglecting their environment, extreme talkativeness about them, megalomania, excessive magical thinking, or being overwhelmed by mystical experiences that could become very frightening, these being pathological aspects of such experiences. In addition, they referred to the costs of being engaged in spirituality, such as hospital admission and long periods of recovery, however enriching the experiences in themselves had been. One of the interviewees expressed this as follows:

> "I never would have wanted to miss this experience. I find it hard. God was with me there, through everything. For years I had the feeling: this is important. But considering what the costs are: not being able to finish my studies, three years of my life, than the question arises: does this divine encounter make up for all the losses? But I see . . . it was truly something very special" (Woman, Protestant; Ouwehand 2020, p. 92)

Only a few people rejected the experiences as only pathological or endorsed an exclusively religious or spiritual explanatory model. Most participants expressed themselves in religious and medical terms in their narratives.

The search for meaning of the experiences and for disentanglement of genuine and pathological aspects thereof were often closely connected to mood swings, characteristic for the course of bipolar disorder, in both studies. One woman states:

> "If you are stable, then you look back on depression and regard sin as something that has been made up. But when you are depressed, then you are afraid that the

> devil really exists". (Woman, raised in a fundamentalist Protestant household, interested in New Spirituality; Ouwehand 2020, p. 105)

Another participant, Muslim, after having found a spiritual path in Sufism during a manic episode, started to doubt his mystical experiences seriously during a depression after this episode and kept a distance from any kind of religiosity for about a year (Ouwehand 2020, p. 105).

Sometimes the quest for meaning took place in the period leading to mania; in other cases, it started after having had religious experiences during mania, and it could take intense forms. It involved reading spiritual literature or searching the internet as a source of information, such as the teachings of Eckhart Tolle, who was mentioned by several participants. One participant says:

> "[I explored] lots of articles and videos, blogs. I think that if I didn't have the internet, I probably would think that I'm crazy. Or I would have just gone with what the doctors told me. [...] And there's a lot of books on the topic and about the subtopics, so being able to spend a whole summer reading about it has been really helpful". (Van der Tempel 2022, p. 95)

Others attended conferences like Crazy Wise, where the healing potential of psychosis is explored, or New Wine, which focuses on Christian healing practices, or intensive personality training like Landmark or Psycho-Synthesis. Others went to peer group meetings of the patient organization or consulted clergy, mental health professionals, or alternative healers. The use of various sources to explain and give meaning to religious experiences was found in both the Dutch and the Canadian study.

The existential experiences of despair, loneliness, and suffering of depressive episodes were generally not described or interpreted in religious terms, with the exception of some Christian believers who experienced the absence of God. In many persons, serious doubts arose regarding former religious interpretations, or the experiences that had lost their significance. Some adopted a medical explanatory model for their experiences during depression and temporarily kept distant from them or from faith or spirituality in general. This interconnectedness of mood, experience, and interpretation made the search for meaning a challenging endeavor for some of the participants. A young Evangelical man reported how difficult it was for him to get grip on the interpretation process with its changing views on self, God, and the world. Even during the interview itself, he knew his story could have been different if he would not have been recovered (Ouwehand 2020, p. 105).

Experienced patients, those with a long history of bipolar disorder, were often better able to put things into perspective about the value and meaning of their religious or spiritual experiences. They had weighed up the costs (hospital admission, long periods of recovery) of their experiences against the benefits thereof. Some pointed to the importance of staying 'grounded' in spirituality. By 'grounded' they meant that spirituality should not be kept separate from ordinary daily reality and that it encompasses much more than just the experiential side. Some older patients judged their immersion in religious experiences and their feverish search for meaning afterwards, such as sins of youth, like "Icarus, flying too close to the sun" (Ouwehand 2020, p. 106), as one man said, although they would not have wanted to have missed them.

The study of Van der Tempel (2022, Canada) showed, on the one hand, similar interpretation processes, like the influence of fluctuations in mood, whereby (hypo)mania reinforced religious appraisals and depression undermined them. During stable periods, more balanced perspectives were developed, sometimes reinforced by religious experiences during such periods. On the other hand, one of Van der Tempel's findings was a more pronounced conflict between rational and scientific values belonging to the atheist worldview and the religious experiences that were difficult or impossible to explain within this frame of reference. One woman reflects:

> "And again, there are moments where logically or scientifically I saw this is bogus, this is garbage, you were in a really bad state. [...] So it's very much been a wrestle between the material and the spiritual understanding of the experience for me". (Van der Tempel 2022, p. 105)

This conflict generated significant uncertainty and self-stigma in participants. Conceptual approaches that helped them in their interpretation process were a social constructionist perspective, which facilitated the holding of plural meanings, suspension of the need for complete and logically consistent explanations, which gave room for greater tolerance of uncertainty, ambiguity, and paradox, and a focus on ethics and moral behavior that de-emphasized the importance of identifying objective truths. One woman, for example, explains:

> "I kind of think of it as Flatland. Like maybe we're just not built yet to understand the universe. Like there are other dimensions that our brains just can't comprehend, and so, in terms of having answers . . . I used to look for answers, and now I'm very happy with just having had my experience and being with the questions . . .". (Van der Tempel 2022, pp. 106/107)

Some participants found a new religious orientation and the majority embraced a form of agnosticism that leaned toward a religious or spiritual perspective in the study by Van der Tempel (2022). As in the study by Ouwehand and colleagues (Ouwehand et al. 2019b), participants generally did not endorse a theoretically consistent explanatory model to account for their religious experiences, but incorporated concepts from various secular/scientific frameworks alongside religious or spiritual explanatory models. The both religious and pathological explanatory model was common in both qualitative studies.

An example of a quest for meaning of religious experiences related to bipolar disorder is described in a detailed case study from the perspective of Dialogical Self Theory, developed by Hubert Hermans (Ouwehand et al. 2020b). 'Peter', who was raised as an atheist and converted to Christianity later in life due to his religious experiences during (hypo)manic episodes, shows the complexity of the process of interpretation over the years and the struggle of a person to find a more balanced attitude toward his religious experiences. Consistent self-reflection (in several documents about his life with bipolar disorder), dialogue with relatives and friends, and psychotherapy eventually led to a more balanced attitude toward faith. This was accompanied by a transition from an outpatient mental health care setting to regular appointments with his general practitioner.

### 2.3. Summarised Findings of the Quantitative Study (Ouwehand 2020)

2.3.1. Religious Experiences and Associations with Diagnosis and Religious Variables

The strength of qualitative studies like those mentioned above is that they give insight into the lived experience and process of interpretation of religious experiences over time, in retrospect, from a patient perspective. The weakness of such studies is that usually only people who have an extraordinary interest in the topic will participate in them. Therefore, results cannot be generalized. For clinical practice, it is also useful to know how often such experiences and explanatory models occur in the population of people with bipolar disorder. Therefore, various types of religious experiences emerging from the interviews were included in a survey (Ouwehand et al. 2019a), using descriptions from qualitative interviews (Table 1). The study was conducted in a specialist outpatient center for bipolar disorder in The Netherlands (n = 196).

Experiences of horizontal transcendence occurred most frequently. Such experiences do not necessarily refer to a supernatural reality or God, but, for example, to human connectedness, to a profound aesthetic experience, or to a deeper dimension of reality than we have in daily consciousness. 'An intense experience of joy, love, beauty or freedom', occurred in 77% of the sample. Those that occurred least often were apparitions of divine beings (21%), 'the experience of being an important religious person' (20%), and religious voices (12%).

**Table 1.** The prevalence of religious or spiritual experiences, as occurring during mania, and without or with a lasting influence (all self-assessed) in a sample for specialized care for bipolar disorder of Altrecht, The Netherlands, N = 196 [1].

| Sample Questions | Yes % | During Mania % | With a Lasting Influence % |
|---|---|---|---|
| 1. An intense experience of happiness, love, peace, beauty or freedom | 77 | 66 | 36 |
| 2. An experience of meaningful synchronicity | 66 | 77 | 25 |
| 3. An intense experience of unity | 57 | 66 | 28 |
| 4. The feeling of having a mission in or for the world | 51 | 77 | 17 |
| 5. An intense experience of the presence of the Divine, of God, or Light? | 44 | 76 | 22 |
| 6. Have you ever had a sudden profound spiritual insight or a sudden revelation or vision? | 37 | 67 | 17 |
| 7. Have you ever seen a religious or spiritual apparition? Of whom? A benevolent spiritual being An evil spiritual being | 21 16 4 | 55 | 11 |
| 8. The feeling of being an important relgious person? | 20 | 85 | 4 |
| 9. Have you ever heard a divine voice speaking to you? Of whom? A benevolent spiritual being An evil spiritual being | 12 9 2 | 54 | 8 |
| | Yes % | During Depression % | With a Lasting Influence % |
| 10. Have you ever experienced a period in which spirituality or faith were completely absent? | 44 | 63 | 10 |
| 11. Have you ever expeerienced a perido of complete absence of the Divine, God or Light? | 36 | 68 | 8 |

[1] For the exact formulation of sample questions, see (Ouwehand et al. 2019a).

Participants marked their religious experiences as occurring significantly more often during manic episodes (see Table 1) than during depression or being stable. Only the experiences of absence of spirituality or faith and the absence of the God or the Divine were, as expected, most often marked as occurring during depression. This result of self-reported episodes of occurrence of the experiences corresponded with the finding that religious experiences occurred significantly more often in patients with a bipolar I diagnosis than with a bipolar II diagnosis (marked by therapists). Of the whole sample, 70% were assessed as having bipolar I disorder by their therapists, with 26% as having a bipolar II disorder. The lasting influence participants attributed to their experiences (see Table 1, last column) ranged from one fifth of the persons who had experienced 'being an important religious person', about half who had 'experiences of horizontal transcendence, presence of God, mystical experiences, sudden spiritual insight, revelation or vision', to two thirds who experienced hearing divine voices. This variation in evaluating long term impact of the experiences supports the findings in the qualitative part of the study that patients, when they are stable, weigh positive and negative aspects of their experiences. For example, some of the interviewees relativized or made jokes about their overly megalomaniac aspirations during mania, although the underlying value of, for example, doing good or spreading love in the world could still be important to them.

Religious or spiritual experiences occurred significantly more often in patients with a present religious affiliation than in persons without a religious affiliation, and more often in persons who were raised in a faith tradition than in persons who did not come from religious homes. Religious experiences also occurred significantly more often in the groups that characterized themselves as being 'only spiritual' or 'religious and spiritual', than in

the 'only religious' or 'neither religious nor spiritual' group in regard to the self-definitions used in the sociology of religion (Bernts and Berghuijs 2016; Possamai 2005).

### 2.3.2. Explanatory Models

The results of the qualitative part of the Dutch study were nuanced in the quantitative part (Ouwehand et al. 2020a). In the interviews, a majority considered their experiences as part of their spiritual development but reflected on the pathological aspects of them as well. Only a few participants viewed their experiences as exclusively religious or exclusively pathological. In the quantitative study (n = 196), half of the participants who had had religious experiences understood their experiences as part of their 'spiritual development' (46%) or considered their experiences as 'both religious and pathological' (42%). The frequencies of the different explanatory models are shown in Table 2.

**Table 2.** Frequencies of types of explanations of religious and spiritual experiences in a Dutch bipolar outpatient sample (retrieved from Ouwehand et al. 2020a).

| Interpretation/Reaction | N | Yes % | No % | Don't Know % |
|---|---|---|---|---|
| 1. They belong to my spritirual development, have deepened my faith | 125 | 46 | 38 | 16 |
| 2. Such experiences have both religious/spiritual and pathological ('ill') features | 124 | 42 | 33 | 25 |
| 3. I keep my distance from such experiences | 121 | 31 | 53 | 16 |
| 4. I am not sure whether they are authentic ('real') religious experiences orbelong to bipolar disorder | 125 | 30 | 53 | 17 |
| 5. Such experiences belong exclusively to my illness | 123 | 15 | 63 | 22 |
| 6. Such experiences are in fact a sign of spiritual crisis or crisis of faith | 124 | 10 | 70 | 20 |
| 7. It is better for me to keep distance from faith or spirituality altogether because such experiences originate from my illness | 124 | 4 | 81 | 15 |

For 15%, such experiences were a sign of bipolar disorder, and one third of the sample doubted the interpretation. In the qualitative study, doubt was an important theme in many interviews, especially during depression, when all former interpretations could be in jeopardy.

Analysis of the relation between various explanatory models and religious and diagnostic variables showed significant associations for religious coping, intrinsic religiosity, religious practice, and self-definition as religious and spiritual, or in reverse, neither religious nor spiritual. This means that the stronger the religious coping and the religious or spiritual involvement of the participants was, the greater their belief in religious experiences as an integral part of their religiosity and spiritual growth and the weaker their conviction that religious experiences are exclusively pathological and better to avoid. No significant associations were found between the various explanatory models and the diagnosis bipolar I or II disorder.

### 2.3.3. Communication and Treatment Expectations

Communication about religious experiences can be challenging for people with bipolar disorder. In the interviews in both studies, participants often reported they were reluctant to share them, and many reported negative experiences in mental health care when having done so. Some people mentioned the wish to cherish their experiences, and their fear that disclosing them to others might imply a loss. Relatives and mental health care professionals were often anxious that increasing interest in spiritual affairs would be a sign of developing mania, they said. Van der Tempel (2022) points to a dual stigma for the atheist group in his study: on top of a sceptic attitude of therapists toward religious experiences, when they are related to a mental disorder, social stigma, due to the atheist context patients live in, can play a role as well. One of the interviewees tells:

"But then there's also my other friends, who—most of my atheist friends—who are just like, 'okay, I don't believe that it's supernatural and maybe you need to take more medication'. [...] But some of the time it's people who are like, 'oh, that's bullshit'. Or 'that's just a coincidence'. Or you know, 'you're just pulling my leg'. They think that I'm messing with them". (Van der Tempel 2022, p. 124)

Differences in religious or spiritual affiliation between patient and professional were an obstacle for some participants; when a religious worldview is absent, such experiences can only be interpreted within a biomedical explanatory model. However, in the quantitative study, 71% had shared their experiences with relatives or friends and almost half with a mental health professional. Only 16% had shared their experiences with clergy or hospital chaplains, and 14% with peers. More than half of the persons with religious experiences (56%) valued conversation with mental health professionals about the topic positively as a future treatment intervention. A remarkable finding was that half of the whole sample considered meaning-making, spirituality, and faith as important topics to address. This percentage was higher for people with a religious or spiritual affiliation (82%) than for the group without affiliation (41%). Formal religious affiliation, however, is not the most important indicator for the degree of religious involvement of a person. Of the group that defined themselves as 'neither religious nor spiritual', about one fifth (18%) considered spirituality or worldview an important topic in treatment. This was 75% for the groups 'only spiritual' and 'religious and spiritual', and half of the group 'only religious' (52%). This last group amounted to only 8% of the total sample.

*2.4. Discussion*

2.4.1. Religious Experiences in Bipolar Disorder: How Crazy or Wise Are They?

The aim of the two studies discussed above was to map the occurrence and variation of religious experiences of persons with bipolar disorder and explore the ways they interpret such experiences when they are stable (Ouwehand 2020; Van der Tempel 2022). Some of the experiences found in the former study have a prevalence that is similar to the general Dutch population, according to sociological studies (Berghuijs 2016, 2017; Bernts and Berghuijs 2016; De Hart 2014). The experience that everything is connected and that nothing happens coincidently, for example, occurs in 53–55% of the general population. In the study by Ouwehand and colleagues (Ouwehand et al. 2019a, n = 196), this was 66%. The experience of the presence of God occurs in 32–50% of the general population and in 44% of the participants of the study. Paranormal experiences, such as contact with deceased and out-of-body or near-death-experiences, reported as an answer to the open question in the survey, are reported in studies in the general population as well. This points to the conclusion that both people with or without a psychiatric diagnosis use language from the same religious and spiritual heritage. Van der Tempel (2022) goes one step further in his conclusion that the phenomenology of religious experiences (especially mystical experiences) of the group of atheists with bipolar disorder in his study was indistinguishable from those of religious individuals with BD, and largely overlapped with those of healthy atheists and religious individuals. Other experiences, such as being an important religious person, occurred less frequently and were retrospectively less often evaluated as having lasting influence on people's lives (Ouwehand et al. 2019a). This experience is probably closer to megalomania or a delusion of grandeur, in psychiatric terms, but will undoubtedly gain less cultural and social approval as well when compared to the more popular mystical experiences.

Some of the participants considered their experiences as an integral part of their spiritual development, others as a transient event. The more important faith or spirituality is for people, the more they integrate religious experiences in their faith or spirituality. More than half of the participants in the quantitative part of the study believed their lives had changed positively (Ouwehand et al. 2019a). For the participants in the qualitative study by Van der Tempel, cognitive dissonance between religious experiences and illness acceptance was common. This was linked to difficulty reconciling spiritual/religious and psychopathological (bio-medical and psychological) explanatory models. The latter were

predominant at the initial stage of the illness, due to the atheist worldview participants endorsed at that time.

According to a Swiss study (Huguelet et al. 2016), the importance of spirituality (measured according to the extent of perceived meaning through faith or spirituality) for persons with schizophrenia is more important than for persons with bipolar disorder. For 41% of the participants with schizophrenia, spirituality was very important, compared to 6% for the group with a diagnosis of bipolar disorder. This outcome is not in line with the findings in Ouwehand and colleagues (Ouwehand et al. 2020a).

Grover and colleagues (Grover et al. 2016a, India) found, in a study of non-medical explanatory models for bipolar disorder in India, that almost half (45%) of the participants (62% Hindu, 34% Sikh, 2% atheist) attributed a religious or supernatural cause to the illness (for example sorcery, evil spirits or possession, divine wrath, planetary influences, or karma). However, only 9% had stopped their medication and about half considered their faith as supportive (59%) or received support from their religious community. Patients with religious explanatory models in the study consulted healers and gurus significantly more often or made use of homoeopathy or ayurveda medicine. In a culture wherein religion is omnipresent and permeates daily life, a medical and a religious explanatory model probably coexist in a non-conflicting way for patients.

Brett and colleagues (Brett 2010; Brett et al. 2014, UK) studied anomalous experiences in not-diagnosed and diagnosed groups with a vulnerability for psychosis. The post-hoc appraisals of the experiences, rather than the content of them, were a stress factor in the interpreting process. Predictive factors for less perceived stress were 'spiritual evaluation' and 'understanding and support of the environment' in this study. These findings concurred with the results from Ouwehand (2020) and point to the hypothesis that a religious or spiritual frame of reference and recognition of the environment are important for both people who have a bipolar disorder and those with a psychotic vulnerability. For atheists, the discrepancy between their worldview and their religious experiences related to (hypo)mania will carry an extra burden to come to terms with them (Van der Tempel 2022).

In both studies (Ouwehand et al. 2019a; Van der Tempel 2022), a clear relation between religious experiences and episodes of the illness appeared. They occurred significantly more often in people with bipolar I than in people with a bipolar II diagnosis and were more often experienced during mania than during depression or when stable. Participants in the Dutch study found it hard to describe religious or spiritual experiences in stable periods. From a psychiatric perspective, it is therefore understandable to evaluate the experiences as hallucinations and delusions. Relevant literature often differentiates between (religious) content of hallucinations or delusions, related to someone's religious background or interest, and the form, being an expression of pathology (Mohr and Pfeifer 2009; Sims 2016). From the above-mentioned studies, however, it becomes clear that experiences related to mania can have meaning for patients long after the episode subsided. Some psychiatrists propose, following William James, to take the consequences or 'fruits' of such experiences in daily life (Sims 2016) as a criterion for pathology or mental health. This is concordant with the finding in Ouwehand and colleagues (Ouwehand et al. 2019a), that the lasting influence of different types of experiences are not equally estimated by participants.

### 2.4.2. A Both/and Approach

The results of both studies, intended to map religious or spiritual experiences from a patient perspective, challenge a strictly psychiatric approach of such experiences. The first question is whether form and content can be clearly divided. In a cultural-anthropological approach, such as the one suggested by Kleinman (1988, 1991), the biomedical model is a social construction, next to personal and cultural/religious explanatory models for illness experiences. Treatment enables—at its best—a dialogue about different explanations. In practice, however, they regularly conflict with each other (Mitchell and Romans 2003, New Zealand; Stroppa and Moreira-Almeida 2013, Brazil). Dissatisfaction with regular mental health care may be the reason for the attractiveness of complementary and alternative

therapies to mental health users in Western countries. De Jonge and colleagues (De Jonge et al. 2018), in a large study in 25 countries, found that 5% of users of complementary and alternative medicine (CAM) in high-income countries also received conventional mental health care. This percentage increased with severity levels of the illness: 12% of people visiting a mental health specialist used CAM. For the group of patients with severe mood disorders, this was 14%. In people with severe mental disorders from high-income countries, as many as 80% of persons reporting contact with CAM also received conventional care. In The Netherlands, CAM is much more accepted and integrated in palliative care, especially when compared to mental health care, where the use of CAM is often not addressed in conventional treatment. This may contribute to confusion in patients about the use of medication and can raise obstacles in the treatment relationship.

Cook (2015) criticizes the lack of clear definitions of religious hallucinations and delusions and the omission to specify the philosophical or religious background of patients in much research. Mohr and Pfeifer (2009) propose to remove the concept of 'religious delusion' from clinical practice because of its stigmatizing effect on patients. Religious coping as a psychological explanation for how people with a serious mental health diagnosis deal with their illness is probably a more helpful category for clinical practice. However, the concept does not sufficiently reflect personal explanatory models and the relation of those interpretations with broader underlying religious trends in modern society.

Ultimately, the question of truth in religious experiences is a normative one that cannot be answered by psychiatry alone, but is discussed in religious traditions, theology, and philosophy. In all religious or spiritual traditions or modern trends, transformation—however phrased or named–is important, although the intended direction of transformation may vary widely. However, in a secularized country such as The Netherlands, religious traditions are losing their influence rapidly and psychiatric patients do not always feel accepted in the religious community. Many new spiritualities bloom abundantly, often without critical examination of their effects on persons with psychiatric vulnerabilities. In the Dutch study, participants defined themselves as significantly more often 'only spiritual' or 'religious and spiritual' than the general Dutch population (Ouwehand 2020). In the study by Van der Tempel (2022, Canada), initially atheist patients inclined toward agnostic and spiritual explanatory models for their experiences but had difficulties finding trustworthy contexts to evaluate them. Mental health professionals are mentioned as conversation partners by almost half of the participants (Ouwehand 2020) and can provide a trustworthy context to explore experiences and the influence thereof on illness acceptance and daily life. In this study, participants expected both a respectful attitude toward their religious experiences and a critical sounding board to evaluate them from mental health professionals (Ouwehand et al. 2019b); more than half expressed the wish to discuss their experiences within treatment (Ouwehand et al. 2020a).

## 3. Future Directions for Research and Implications for Clinical Practice

### 3.1. Future Directions for Research

Jackson and colleagues (Jackson et al. 2022), in their recent review study of the literature to date about bipolar disorder and religion or spirituality, concluded that it is difficult to summarize the present state of affairs of scientific knowledge about the intended subject of interest. Studies use different measurements of religiosity, sometimes without being specific, and many studies lack scientific rigor. Of the 400 papers the authors examined, they excluded most studies for to the following reasons: studies in diagnostically heterogeneous samples wherein the data for bipolar patients and the proportion of this diagnostic group were not distinguished from the total sample; studies that did not include effect size and $p$ value for all main results; and studies with a self-reported diagnosis of bipolar disorder of the participants. Some studies did not adequately correct for multiple testing in their statistical analyses or use inappropriate modelling. In the 14 included quantitative studies,

intrinsic religiosity, non-organized (e.g., private) religious activities, and positive religious coping were most consistently associated with beneficial effects. Mosqueiro and colleagues (Mosqueiro et al. 2020), in their short review of the available literature, noticed that religion and spirituality are important for patients with bipolar disorder and that religious coping is a common source of coping across cultures, but that the illness influences religiosity or spirituality as well. Both positive and negative religious coping or religious struggles and their effects on recovery and well-being in bipolar disorder need better qualified research. However, this is not an easy task. Many measures of different aspects of religiosity, such as the brief religious coping scale (Brief RCOPE, Pargament 1999; Pargament et al. 2011, USA), one of the most used measures for research in the domain of religion and health, especially in the USA, feature a Christian bias. The brief RCOPE contains 10 or 14 items that all refer to 'God'. This likely does not correspond with religious expressions of persons who identify themselves as 'only spiritual' in secularized societies, or persons with an agnostic worldview with spiritual features, as in the study by Van der Tempel (2022). The construction of a distinctive concept of 'spirituality' is scientifically not agreed upon, however. Koenig (2018) reviews the most commonly used measures of religiosity and of religious coping in his handbook 'Religion and Mental Health'. His critique of scales that use 'spirituality' as a construct is that they are contaminated by indicators of mental health or wellbeing and therefore inappropriate for studying relationships between religion and health.

Riegel and Unser (2021, Germany) developed a supplement to the RCOPE scale, with secular meaning-focused coping items (Trust in Science, consequences of Lifestyle, and Reappraisal of Science's Power) based on the work of philosopher Charles Taylor. They tested their instrument on university students, and this would probably have been a more appropriate instrument for the highly educated Western populations in the studies by Ouwehand (2020) and Van der Tempel (2022) than the RCOPE. The problem of construing distinctive concepts for measuring a multi-layered and context-dependent concept such as 'religion' for quantitative research will remain a challenge. This, of course, not only applies to the study of religiosity in relation to bipolar disorder but to the whole domain of religion and mental health.

For clinical practice, qualitative research is helpful to gain insight in the enhancing or disruptive role religion or spirituality play in individual lives or groups from a first-person perspective. Qualitative explorations also can serve as steppingstones to observational studies and clinical trials, according to Koenig (2018). Jackson and colleagues (Jackson et al. 2022) applied the same exclusion criteria to the qualitative studies as to the quantitative studies the authors reviewed. Only two qualitative studies based on interviews and two case reports were included in their review. The qualitative studies described in Section 2 of this article were not included by Jackson and colleagues because the authors considered the method of diagnostic confirmation of the self-reported diagnosis of participants (interviews by a hospital chaplain together with a psychiatrist trainee, taking a short psychiatric history, and, in cases of doubt, consultation of the treating psychiatrist) not compliant with their inclusion criteria.

Apart from the question whether a self-reported diagnosis of stable patients cannot serve as an inclusion criterion for a qualitative study, we would like to stress that there is a lack of studies into the lived experiences of patients. Such studies give more insight into the complexity of the role religion or spirituality play in relation to illness. A recent systematic review study into subjective experience and the meaning of delusions and psychosis led to a qualitative evidence synthesis of three themes: 1. A radical re-arrangement of the lived world dominated by intense emotions; 2. Doubting, losing, and finding oneself again within delusional realities; and 3. Searching for meaning, belonging, and coherence beyond mere dysfunction (Ritunnano et al. 2022). Studies were eligible when providing an analysis of lived experience of delusions or predelusional phenomena of individuals with a clinical high risk stage of psychosis or a diagnosable affective or non-affective psychotic disorder (as clinically defined or self-reported); this included persons with bipolar disorder.

This is an example of a study that can have significance especially for the treatment of bipolar I patients and shows that transdiagnostic studies from a first-person perspective can contribute to the body of psychiatric knowledge as well. In general, we must conclude that our knowledge about the relation between bipolar disorder and religion is still limited. Future research would benefit from contextual, multi-disciplinary, and mixed-methods study designs to contribute to the synthesis of knowledge of the complex relationship between religion and bipolar disorder.

*3.2. Implications for Clinical Practice*

The American Psychological Association, the American Psychiatric Association (WPA), and the Royal College of Psychiatrists all have a section concerning Religion and Spirituality. In 2016, the WPA published a position statement (Moreira-Almeida et al. 2016) recommending, among other things, the taking of a religious/spiritual history of patients to assess the influence of belief and practices of patients on the illness and the importance patients attach to this domain in life in relation to their illness. The practice of assessment of religion and spirituality in mental health care, let alone the application of evidence-based interventions in treatment, is still in its infancy, although a growing body of literature shows that interest for the subject is increasing.

Van Nieuw Amerongen-Meeuse and colleagues (Van Nieuw Amerongen-Meeuse et al. 2018, The Netherlands), in a qualitative study, point to the 'religiosity gap', referring to lower rates of religious involvement among mental health care professionals compared with that of patients. Patients can perceive disrespect for their religiosity and misunderstanding or misinterpretation from the part of mental health professionals. However, they can also experience a religiosity match when there is space to share their religiosity. Spiritual care needs of patients are not always explicitly expressed to professionals (Van Nieuw Amerongen-Meeuse et al. 2019, The Netherlands). The quantitative part of that study confirms that a substantial number of mental health patients prefer to address faith or spirituality in treatment, and this goes beyond a referral to a hospital chaplain or taking a spiritual history. Attention to religion and spirituality may benefit the treatment alliance (Van Nieuw Amerongen-Meeuse et al. 2021, The Netherlands).

Vieten and Scammell (2015) present sixteen research-based spiritual and religious competencies for mental health care professionals, encompassing attitudes, knowledge, and skills. They regard an empathic, respectful, and appreciative attitude towards patients' spiritual or secular backgrounds as a necessary condition to address the topic in treatment. Their book contains advice on addressing religious issues in clinical practice. David Rosmarin developed a transdiagnostic clinical intervention, Spiritual Psychotherapy for Inpatients, Residential, and Intensive Treatment (SPIRIT), that benefits both patients with high and low levels of religiosity (Rosmarin et al. 2021, USA). This spiritual psychotherapy was more effective when provided by nonreligious clinicians compared to clinicians with a religious affiliation (Rosmarin et al. 2022, USA). At the moment, this intervention is adapted for the Dutch context (Van Nieuw Amerongen-Meeuse et al. 2024).

The literature about spiritual care provided by hospital chaplaincy in mental health care is increasing as well. Hospital chaplains have a rich experience and substantive knowledge of religious and spiritual traditions, mostly work transdiagostically and inclusively (e.g., in multi-faith teams, Louis and Isakjee 2019, UK), and cooperate with experts-by-experience and recovery colleges (Jeffery and Boyle 2019, UK). Clinical practice can benefit from cooperation between different disciplines in mental health care.

In The Netherlands, a guideline for exploring and implementing meaning-making and spirituality in mental health care is published in May 2023 (GGZ-Standaarden 2023).

This guideline points to three ways in which spirituality and illness are interwoven: 1. Spirituality and mental health mutually influence each other. They can both enhance or impede each other; 2. Mental health problems can be colored spiritually; and 3. Patients may have spiritual explanatory models for illness experiences and spirituality can influence their view on treatment.

Looking at the literature on religion and bipolar disorder specifically, we can trace examples of all three points: 1. Mood swings and course of the illness influence the way patients perceive spirituality (Duckham 2011, USA) and religious or spiritual experiences related to mania. This results in a search for meaning and for many in a struggle to disentangle genuine spirituality from hyper-religiosity (Michalak et al. 2006, Canada; Ouwehand et al. 2019b, The Netherland; Van der Tempel 2022, Canada). During depression, faith and spirituality are often absent, although some patients still may feel some support from their spirituality (Ouwehand et al. 2018, The Netherlands). Patients can also become disappointed in faith or spirituality (Mitchell and Romans 2003, New Zealand). The case study by Duckham (2011, USA) is a rich description of the positive influence of psychoanalytic therapy on self and God-image over the years; 2. A clear example of the spiritual coloring of illness symptoms is the religious or spiritual content of psychotic symptoms (Grover et al. 2016a, India; Hempel et al. 2002, USA; Khan and Sanober 2016, Pakistan; Ouwehand et al. 2018, The Netherlands; Van der Tempel 2022, Canada). This coloring may be dependent on cultural and religious context (Khan and Sanober 2016, Pakistan; Luhrmann 2011); 3. The use of religious or spiritual explanatory models for illness experiences is mentioned in several studies (Granek et al. 2018, Israël; Ouwehand et al. 2019b, Ouwehand 2020, The Netherlands; Van der Tempel 2022, Canada). It can lead to a paradigm conflict between medical and spiritual advisors or disagreement in treatment (Mitchell and Romans 2003, New Zealand; Stroppa and Moreira-Almeida 2013, Brazil), but religious advisors can also pave the way to treatment or taking medication (Granek et al. 2018, Israël). Although no evidence-based treatment protocols are developed yet, all three mentioned topics can be addressed in various phases of assessment and treatment when done so in a respectful manner.

The core recommendations for clinical practice in the above-mentioned Dutch guideline are not specific for bipolar disorder and correspond with the international literature. Attention to spirituality, religion, and meaning-making of patients and relatives is important and can improve the treatment relationship. Patients can be reluctant to address spirituality, so it is important for professionals to take the initiative and subsequently integrate the topic in treatment when patients desire this. Assessment of the specific experiences and beliefs of patients and relatives, their spiritual perspective on illness and health, spiritual needs, religious context, and cultural identity is advisable. A developing body of assessment tools and/or interventions can be deployed (described in the guideline). An open and inviting attitude without judgements about the veracity of patients' beliefs is helpful. A professional can listen carefully, but it is not always necessary to do something; this is because existential and religious questions often do not have or need any cut-and-dried answers. Mental health professionals need to cooperate with other professionals, experts-by-experience, specialized therapists, spiritual counsellors, and/or religious or spiritual organizations, when patients desire this or when professionals run into their own limitations with regard to the subject. Self-reflection about the professional's own attitude toward spirituality and how this attitude may influence therapeutic interventions is a prerequisite for the possibility of addressing the domain of faith or spirituality appropriately and respectfully.

**Funding:** The qualitative part of the research received no external funding. The quantitative part of the research was funded by the Association for Christian Care of Mental and Nervous Diseases-Support Foundation (Stichting tot Steun VCVGZ, 17 March 20217); Han Gerlach Foundation Study Fund (16 January 2017).

**Institutional Review Board Statement:** The study was approved by the Regional Medical Ethical Committee of the University Medical Centre Groningen (METc2014.475) and the Scientific Committee of Altrecht Mental Health Care (2016-40/oz1620).

**Informed Consent Statement:** Informed consent was obtained from all subjects involved in the study.

**Data Availability Statement:** No new data were created or analyzed in this study. Data sharing is not applicable to this article.

**Conflicts of Interest:** The author declares no conflicts of interest.

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
