# Peer review of "Religious Experiences in the Context of Bipolar Disorder: Serious Pathology and/or Genuine Spirituality? A Narrative Review against the Background of the Literature about Bipolar Disorder and Religion"

_religions, doi:10.3390/rel15030274_

Round 1

Reviewer 1 Report

Comments and Suggestions for Authors

I. Substantive issues:

  1. I assess the work positively because it is a valuable presentation of rarely encountered research.
  2. It is important to pay attention to the non-judgmental approach to patients' beliefs.
  3. I appreciate the recognition of the need for collaboration between healthcare workers and other specialists, encouragement for reflection on one's approach to spirituality, and the expectation of treating people's faith with respect.

II. Formal issues: I suggest replacing the abbreviations used in the text with full names. Alternatively, after the first full expression, add the abbreviation. In this case, the use of the abbreviation will be permissible later in the paper. Results considered most important should be presented in the form of a chart.

Reviewer 2 Report

Comments and Suggestions for Authors

The article makes an original and interesting contribution as it deals with the understudied topic of the relationship between religion/spiritual experience and bipolar disorder on which there is no extensive literature. The text is well written and has a good logical and argumentative structure in all sections. Particularly appreciable is the critical review of the existing literature, albeit scarse, cited in the Introduction regarding religious coping, fasting, and religious experience, delusion and hallucinations.

For this reason only some minor revisions are recommended, aimed at further enhancing the research work.

The first suggestion is to report - in section 2 - at least some exemplary excerpts from the qualitative interviews conducted with the patients involved in the study.

The second suggestion is to integrate the reflection on the implications for clinical practice - in section 3 - with the indications elaborated in the literature on “spiritual care” and developed in the field of nursing sciences.

Author Response

please see the attachement
